# Validation study of bullous pemphigoid and pemphigus vulgaris recording in routinely collected electronic primary healthcare records in England

Monica S M Persson ,[1] Karen E Harman,[1] Yana Vinogradova,[2]
Sinead M Langan,[3] Julia Hippisley-Cox,[4] Kim S Thomas ,[1] Sonia Gran[1]

[1]Centre of Evidence Based Dermatology, University of Nottingham, Nottingham, UK
[2]Primary Care, University of Nottingham, Nottingham, UK
[3]Department of Non-communicable Disease Epidemiology, London School of Hygiene and Tropical Medicine, London, UK
[4]Nuffield Department of Primary Care Sciences, University of Oxford, Oxford, UK

**Correspondence to**
Dr Monica S M Persson;
Monica.Persson1@nottingham.ac.uk

## ABSTRACT

**Objectives** The validity of bullous pemphigoid and pemphigus vulgaris recording in routinely collected healthcare data in the UK is unknown. We assessed the positive predictive value (PPV) for bullous pemphigoid and pemphigus vulgaris primary care Read codes in the Clinical Practice Research Datalink (CPRD) using linked inpatient data (Hospital Episode Statistics (HES)) as the diagnostic benchmark.

**Setting** Adult participants with bullous pemphigoid or pemphigus vulgaris registered with HES-linked general practices in England between January 1998 and December 2017. Code-based algorithms were used to identify patients from the CPRD and extract their benchmark blistering disease diagnosis from HES.

**Primary outcome measure** The PPVs of Read codes for bullous pemphigoid and pemphigus vulgaris.

**Results** Of 2468 incident cases of bullous pemphigoid and 431 of pemphigus vulgaris, 797 (32.3%) and 85 (19.7%) patients, respectively, had a hospitalisation record for a blistering disease. The PPV for bullous pemphigoid Read codes was 93.2% (95% CI 91.3% to 94.8%). Of the bullous pemphigoid cases, 3.0% had an HES diagnosis of pemphigus vulgaris and 3.8% of another blistering disease. The PPV for pemphigus vulgaris Read codes was 58.5% (95% CI 48.0% to 68.9%). Of the pemphigus vulgaris cases, 24.7% had an HES diagnosis of bullous pemphigoid and 16.5% of another blistering disease.

**Conclusions** The CPRD can be used to study bullous pemphigoid, but recording of pemphigus vulgaris needs to improve in primary care.

## INTRODUCTION

Bullous pemphigoid and pemphigus vulgaris are rare autoimmune blistering diseases characterised by blistering of the skin and mucous membrane and associated with high mortality.[1 2] There is a continued need to understand the causes, natural history and disease associations in order to better inform patients and plan treatment. However, there is a scarcity of recent epidemiological research on these distinct diseases in the UK and many unanswered questions. Furthermore,

### Strengths and limitations of this study

► This is the first study to validate the recording of bullous pemphigoid and pemphigus vulgaris in primary healthcare records in England.
► The study involved a large sample size and the application of algorithms to identify both primary care and benchmark hospital inpatient diagnoses.
► Blistering diagnoses recorded in hospital inpatient records were regarded as the benchmark diagnosis, but may be subject to misclassification.

interpretation of available evidence is limited by the poor external validity and lack of power that are a feature of hospital-based studies.[3–6]

Routinely collected electronic healthcare records (EHR) may offer an excellent opportunity to examine bullous pemphigoid and pemphigus vulgaris at population level due to the availability of data on a large number of patients that are broadly representative of the UK population.[7] Furthermore, the presence of prescription data allows the important association between possible drug exposure and bullous pemphigoid to be established, and longitudinal follow-up allows long-term outcomes to be assessed.[8] However, as EHRs were created for clinical purposes, and not directly for research, it is essential to establish the validity of the diagnoses of interest.[9]

Diagnoses of bullous pemphigoid or pemphigus vulgaris are usually made in secondary care and entered into the primary care records by general practitioners (GP) or other practice staff using Read codes. Read codes have been used previously to identify patients with bullous pemphigoid and pemphigus vulgaris in EHRs, but have not been validated.[2 8 10 11] A previous study examined the incidence of bullous pemphigoid and pemphigus vulgaris in the UK and found that the average age of patients with

pemphigus vulgaris was older than observed in other studies.[2] Furthermore, it coincided with the peak age for bullous pemphigoid, suggesting that pemphigus vulgaris Read codes may not be accurate and that, in some cases, bullous pemphigoid was misclassified as pemphigus vulgaris.

Due to the linkage between primary and secondary care data, we now have the opportunity to externally validate bullous pemphigoid and pemphigus vulgaris Read codes. This study aimed to determine the positive predictive value (PPV) for bullous pemphigoid and pemphigus vulgaris primary care diagnostic Read codes in Clinical Practice Research Datalink (CPRD) using linked inpatient data (Hospital Episode Statistics; HES) as the benchmark.

## METHODS

### Study design

This was a cohort study to validate Read codes for bullous pemphigoid and pemphigoid vulgaris in the CPRD. We followed the guidelines for the reporting of studies conducted using observational routinely collected health data.[12]

### Data sources

#### Clinical Practice Research Datalink

The CPRD GOLD is a longitudinal database of UK general practices using the Vision software system. It contains anonymised healthcare records from 1987 onwards for approximately 17 million patients with a current coverage of approximately 2.7 million (4%) of the UK population. Routinely collected data within the CPRD include demographic and clinical information.[7] Symptoms and diagnoses, generated from primary care consultations or obtained from hospital discharge or specialist clinic letters, are entered by GPs or other practice staff using Read codes.[13]

The data in the CPRD have repeatedly been shown to be of good research quality.[14] At a practice level, participating practices are assigned an 'up-to-standard' date on completion of regular audits confirming data quality. At the patient level, records are assessed and patients are deemed 'acceptable' if data checks indicate that their record meets prespecified quality standards.

#### Hospital Episode Statistics

HES admitted patient care data are available for approximately 75% of English practices, covering over 10 million patients, that have consented to provide patient-level information from linked resources.[15] Linkage is based on each patient's National Health Service number, which is unique and remains unchanged through their lifetime, along with other identifiers (eg, gender, date of birth, postcode).[16] HES admitted patient care contains demographic and clinical data, coded using the International Classification of Diseases Version 10 (ICD-10).[17 18] Each hospital episode is attributed one primary diagnosis, while comorbidities and other diagnoses are recorded as secondary diagnoses.

### Study population

Adult men and women registered with 410 HES-linked general practices in England during the period of 1 January 1998 to 31 December 2017 were included.

### Case definition of bullous pemphigoid and pemphigus vulgaris

Diagnostic Read code lists for bullous pemphigoid and pemphigus vulgaris were developed based on the expert opinion of two consultant dermatologists (KEH and SML) and a GP (JHC). Bullous pemphigoid Read codes were classed as specific (M145000, 'bullous pemphigoid') or broad (M145.00, 'pemphigoid'; M145z00, 'pemphigoid not otherwise specified'). Likewise, pemphigus vulgaris codes were grouped as specific (M144600, 'pemphigus vulgaris'; M144500, 'pemphigus vegetans') or broad (M144.00, 'pemphigus'; M144z00, 'pemphigus not otherwise specified').

Clinical data were sought for all patients who had at least one specific or broad Read code for bullous pemphigoid or pemphigus vulgaris. Based on clinical experience, it was considered inappropriate to define a case based on the first recording of bullous pemphigoid or pemphigus vulgaris without taking into account subsequent Read code entries for blistering diseases which were sometimes multiple and conflicting. A code-based algorithm (figure 1) was therefore applied to determine the most likely diagnosis. The date of diagnosis was taken as the first recording of the Read code generated through the algorithm. Patients were considered cases if their most likely blistering disease diagnosis, generated through the algorithm, was bullous pemphigoid or pemphigus vulgaris. Those with diagnoses for other blistering diseases were excluded. Further information about the blistering disease codes used is available in online supplementary table 1.

### Observation period

The observation period commenced on the latest of (1) 1 January 1998, (2) the patient's 18th birthday, (3) 1 year after the patient was registered with their current general practice, or (4) the practice's up-to-standard date. A 1 year lag period from the patient's registration date with their GP was imposed to minimise the risk of prevalent cases being identified as incident cases.[19]

The observation period terminated on the earliest of (1) 31 December 2017, (2) the date of death, (3) the date the patient left the practice, (4) the practice's last data collection date, and (5) the most recent linkage date between CPRD and HES.

### Validation of bullous pemphigoid and pemphigus vulgaris Read codes

For people with a bullous pemphigoid or pemphigus vulgaris diagnosis in primary care, we identified those with a hospital inpatient episode recording a blistering disease (ICD-10: L10, L12, L13) as a primary or secondary

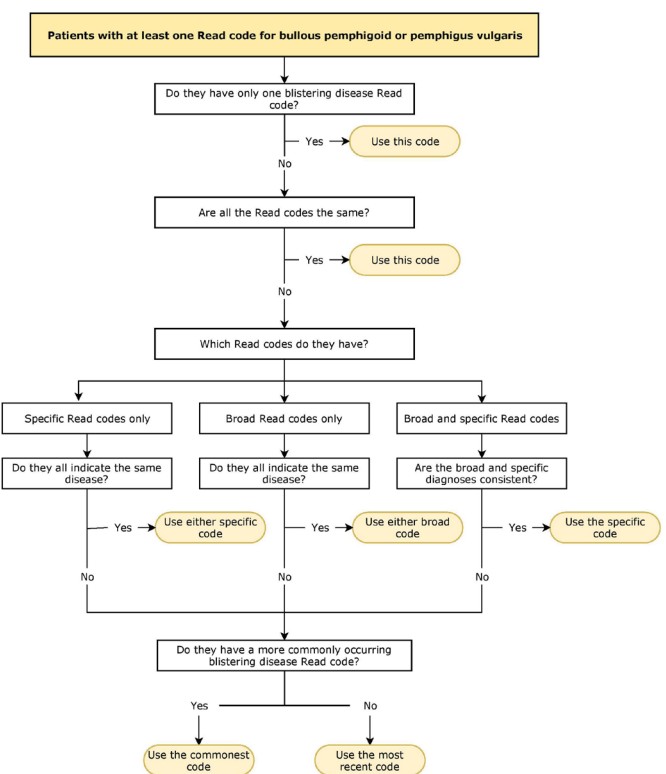

**Figure 1** The code-based algorithm used to determine a diagnosis of bullous pemphigoid or pemphigus vulgaris using Read codes in the Clinical Practice Research Datalink (CPRD). The date of diagnosis was taken as the first recording of the Read code generated through the algorithm.

diagnosis at any point. Due to the presence of multiple records per patient, an algorithm based on clinical knowledge (figure 2) was applied to determine one secondary care ICD-10 blistering disease diagnosis per patient, which was regarded as their benchmark diagnosis. The date of diagnosis was taken as the first recording of the diagnosis generated through the algorithm. The distribution of the benchmark diagnoses was examined for bullous pemphigoid and pemphigus vulgaris cases identified in the CPRD.

ICD-10 codes L12.0 (bullous pemphigoid) and L12.9 (pemphigoid, unspecified) were classed as 'bullous pemphigoid'. ICD-10 codes L10.0 (pemphigus vulgaris), L10.1 (pemphigus vegetans) and L10.9 (pemphigus, unspecified) were classed as 'pemphigus vulgaris'. All other blistering disease ICD-10 codes and where the HES algorithm generated an uncertain diagnosis were classed as 'other'. See online supplementary table 1 for the full list of blistering disease codes.

### Statistical analysis
The sex and age of bullous pemphigoid and pemphigus vulgaris cases were presented descriptively. Age was assessed for normality visually, the mean (SD) or median (IQR) was calculated as appropriate. For bullous pemphigoid and pemphigus vulgaris cases with a hospital inpatient record for a blistering disease, the PPV and associated 95% CI were calculated for the specific, broad

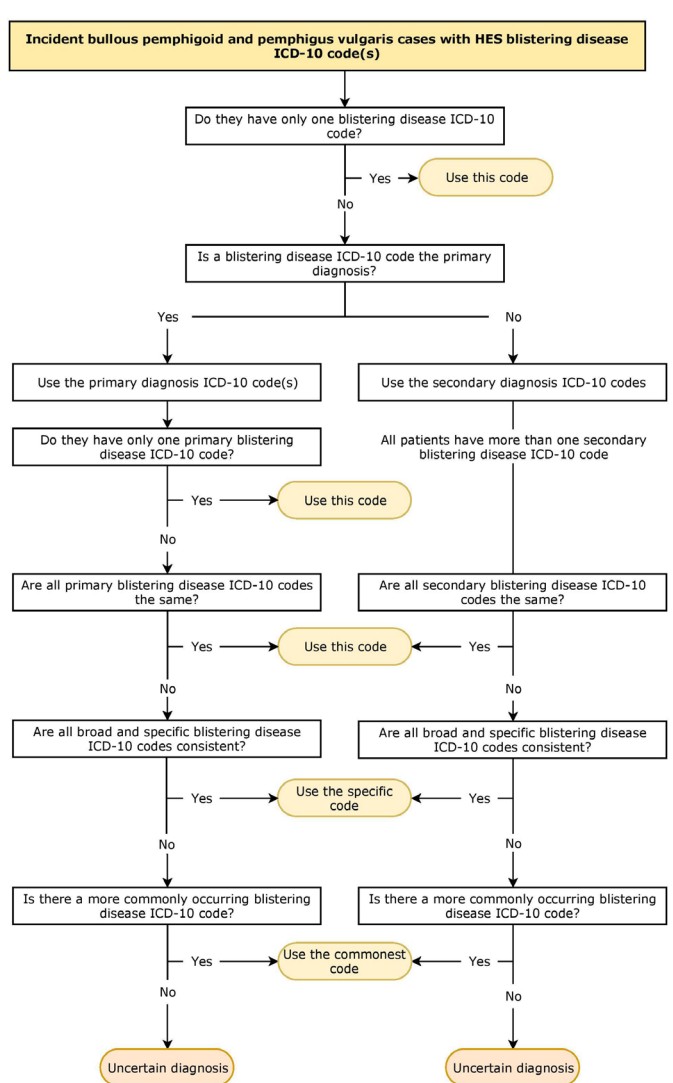

**Figure 2** Code-based algorithm used to determine a benchmark diagnosis from secondary care HES data in patients identified from Clinical Practice Research Datalink (CPRD) as having bullous pemphigoid or pemphigus vulgaris. The date of diagnosis was taken as the first recording of the diagnosis generated through the algorithm. HES, Hospital Episode Statistics; ICD-10, International Classification of Diseases Version 10.

and combined Read codes for bullous pemphigoid and pemphigus vulgaris against their benchmark diagnosis from HES. The blistering disease secondary care diagnoses were presented descriptively. Analyses were conducted using Stata V.16 (StataCorp).

### Sensitivity analysis
The most recent Read code was used for patients with conflicting Read codes (ie, patients who did not have a more commonly occurring Read code; last step of algorithm, figure 1). This was done under the assumption that diagnoses are likely to be refined over time and the most recent code was likely to be most accurate. The impact of this assumption was investigated through a sensitivity analysis comparing the PPVs for bullous pemphigoid and

 3

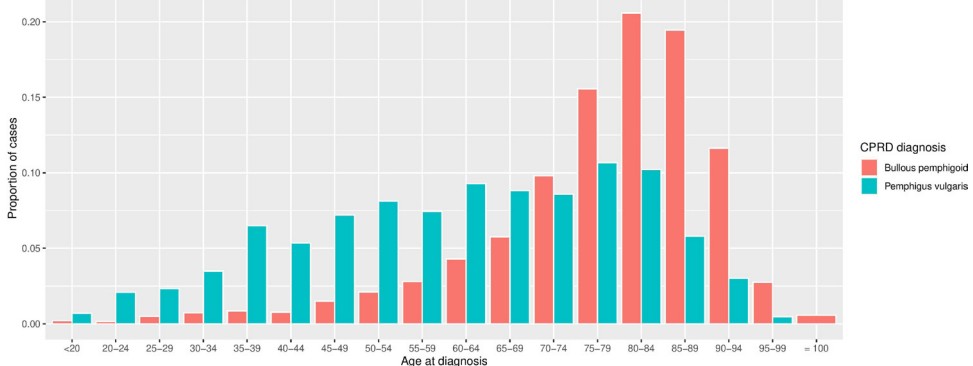

**Figure 3** Age at diagnosis (years) in patients with bullous pemphigoid (n=2468) and pemphigus vulgaris (n=431). A peak is observed at 80–84 years for bullous pemphigoid recording, while two peaks are seen for pemphigus vulgaris; one at 50–54 and a later, and slightly higher, peak at 75–79 years. CPRD, Clinical Practice Research Datalink.

pemphigus vulgaris in all cases versus excluding patients with conflicting Read codes.

### Patient and public involvement
The study was supported by a patient and public advisory group which provided input to the programme of research. We met with this advisory group on a regular basis for the duration of the study. The advisory group commented on the design of the study and, at the end of the study, they commented on the findings and suggested how to share the findings with the public.

## RESULTS
### Study population
A total of 2899 incident cases of bullous pemphigoid and pemphigus vulgaris were identified in HES-linked practices using the algorithm (see online supplementary figure 1 for patient flow diagram). The median (IQR) age at diagnosis was 81 years (73–87) for the 2468 bullous pemphigoid cases, and 64 years (48–77) for the 431 pemphigus vulgaris cases. The peak age for bullous pemphigoid recording was around 80–84 years. In contrast, two peaks were observed for pemphigus vulgaris case recording, one at 50–54 years and a later, slightly higher, peak at 75–79 years (figure 3).

The proportion of women was 56.7% and 64.5% for bullous pemphigoid and pemphigus vulgaris, respectively. Of the 2899 cases, 223 (9.0%) had at least one conflicting Read code indicative of another blistering disease or a general blistering disease code (eg, 'bullous dermatoses').

### Validation of bullous pemphigoid and pemphigus vulgaris diagnostic codes
Of 2899 incident cases of bullous pemphigoid or pemphigus vulgaris, 882 (30.4%) had at least one hospital episode recording a blistering disease diagnosis. In 44.0% of these patients, it was a primary diagnosis. The remaining 2017 (69.6%) patients did not have any hospitalisation records listing a blistering disease diagnosis and were not analysed in the validation study.

Using the algorithm, 10 patients (1.1%) had an uncertain HES diagnosis and were classified as 'other'. The blistering disease codes of these cases are shown in online supplementary table 2. The distribution of HES diagnoses for the bullous pemphigoid and pemphigus vulgaris incident cases and PPV for bullous pemphigoid and pemphigus vulgaris Read codes are shown in table 1.

### Bullous pemphigoid
Of 2468 incident cases, 797 (32.2%) had a hospitalisation record for a blistering disease. Cases were most commonly coded with specific (69.1%) rather than broad (30.8%) Read codes. The PPV of the bullous pemphigoid Read codes combined was 93.2% (95% CI 91.3% to 94.8%). Broad Read codes had a lower PPV (87.8%, 95% CI 83.1% to 91.4%) than the specific codes (95.6%, 95% CI 93.6% to 97.1%).

### Pemphigus vulgaris
Of 431 incident cases, 85 (19.7%) had a hospitalisation record for a blistering disease. Cases were more commonly coded with broad disease codes (74.1%) than specific codes (25.9%). The PPV of the pemphigus vulgaris Read codes combined was 58.8% (95% CI 48.0% to 68.9%). Specific Read codes showed a higher PPV (81.8%, 95% CI 58.8% to 93.4%) than broad codes (50.8%, 95% CI 38.4% to 63.1%).

### Sensitivity analysis
The sensitivity analyses excluding the 223 patients with conflicting Read codes resulted in broadly unchanged PPVs. The PPV for the 776 patients with consistent bullous pemphigoid Read codes was 93.6% (95% CI 91.6% to 95.1%). The PPV for the 75 patients with consistent pemphigus vulgaris Read codes was 61.3% (95% CI 49.7% to 71.8%).

### Inconsistent codes
#### Bullous pemphigoid
A benchmark, secondary care diagnosis of pemphigus vulgaris or another blistering disease diagnosis was made for 24 (3.0%) and 30 (3.8%) of the 797 cases, respectively.

**Table 1** Distribution of benchmark blistering disease diagnoses for bullous pemphigoid and pemphigus vulgaris incident cases identified in the CPRD, shown with the PPV (95% CI) for specific, broad and combined Read codes for each disease

| CPRD Read code | Hospital episode diagnosis (benchmark) | | | | |
| | Bullous pemphigoid | Pemphigus vulgaris | Other | n | % PPV (95% CI) |
| --- | --- | --- | --- | --- | --- |
| **Bullous pemphigoid** | | | | | |
| Specific (M145000) | 527 (95.6%) | 12 (2.2%) | 12 (2.2%) | 551 | 95.6 (93.6 to 97.1) |
| Broad (M145.00, M145z00) | 216 (87.8%) | 12 (4.9%) | 18 (7.3%) | 246 | 87.8 (83.1 to 91.4) |
| Total | 743 (93.2%) | 24 (3.0%) | 30 (3.8%) | 797 | 93.2 (91.3 to 94.8) |
| **Pemphigus vulgaris** | | | | | |
| Specific (M144600, M144500) | 2 (9.1%) | 18 (81.8%) | 2 (9.1%) | 22 | 81.8 (58.8 to 93.4) |
| Broad (M144.00, M144z00) | 19 (30.2%) | 32 (50.8%) | 12 (19.1%) | 63 | 50.8 (38.4 to 63.1) |
| Total | 21 (24.7%) | 50 (58.8%) | 14 (16.5%) | 85 | 58.8 (48.0 to 68.9) |

ICD-10 codes L10.0 (pemphigus vulgaris), L10.1 (pemphigus vegetans) and L10.9 (pemphigus, unspecified) are classed as a hospital episode diagnosis of 'pemphigus vulgaris'. ICD-10 codes L12.0 (bullous pemphigoid) and L12.9 (pemphigoid, unspecified) are classed as 'bullous pemphigoid'. All other blistering disease ICD-10 codes and where the HES algorithm generated an uncertain diagnosis (n=7 for bullous pemphigoid and n=3 for pemphigus vulgaris) are classed as 'other'.
CPRD, Clinical Practice Research Datalink; HES, Hospital Episode Statistics; ICD-10, International Classification of Diseases Version 10; PPV, positive predictive value.

The most common 'other' benchmark diagnosis was cicatricial pemphigoid (n=11, 36.7%; L12.1, mucous membrane pemphigoid). Other benchmark diagnoses were other pemphigus (L10.8), other pemphigoid (L12.8) and bullous disorder unspecified (L13.9).

### Pemphigus vulgaris
Of the 85 cases with a primary care diagnosis for pemphigus vulgaris, 21 (24.7%) had an HES diagnosis for bullous pemphigoid and 14 (16.5%) for another blistering diagnosis. The most common 'other' secondary care diagnosis was for pemphigus foliaceus (n=6, 42.9%; L10.2). Other benchmark diagnoses were drug-induced pemphigus (L10.5), cicatricial pemphigoid (L12.1), dermatitis herpetiformis (L13.0), subcorneal pustular dermatitis (L13.1) and other specified bullous disorder (L13.8).

### DISCUSSION
### Main findings
Incident diagnoses of bullous pemphigoid and pemphigus vulgaris identified in the CPRD were compared with the blistering disease diagnoses made in secondary care (benchmark diagnosis). Diagnoses were recorded in the CPRD using specific (M145000, 'bullous pemphigoid'; M144600, 'pemphigus vulgaris'; M144500, 'pemphigus vegetans') or broad (M145.00, 'pemphigoid'; M145z00, 'pemphigoid not otherwise specified'; M144.00, 'pemphigus'; M144z00, 'pemphigus not otherwise specified') Read codes. Overall, 882 (30.4%) patients identified in the CPRD also had hospitalisation records for a blistering disease. We found good consistency between

primary and secondary care diagnoses for bullous pemphigoid and a high PPV for bullous pemphigoid Read codes (93.2%). The PPV for the specific Read codes for bullous pemphigoid was higher (95.6%) than that of the broad Read codes (87.8%) and captured a greater proportion of cases (69.1%). The consistency was not as good for pemphigus vulgaris, with a moderate PPV for pemphigus vulgaris Read codes (58.8%). The majority of cases of pemphigus vulgaris were coded with broad disease codes (74.1%) with lower PPVs (50.8%). The PPV for the specific pemphigus vulgaris Read codes was higher (81.8%), but captured only a subset of the population (25.9%).

Our study shows that the codes for bullous pemphigoid in the CPRD appear to be capturing people who have the disease and cases reflect expected age distributions. Approximately 93% of bullous pemphigoid cases identified using the algorithm are likely to have bullous pemphigoid. In contrast, only 59% of pemphigus vulgaris cases identified are likely to have pemphigus vulgaris, with almost 25% probably suffering from bullous pemphigoid. The probable misclassification of bullous pemphigoid cases as pemphigus vulgaris likely explains the peak of pemphigus vulgaris codes observed at 75–79 years, coinciding with the peak age for bullous pemphigoid. Consequently, the use of data from the CPRD for pemphigus vulgaris research may be limited.

### Comparability with other studies
To date, no studies have validated Read codes and only two studies have validated ICD codes for bullous pemphigoid or pemphigus vulgaris using EHRs (table 2). The

**Table 2** Published validation studies of bullous pemphigoid or pemphigus vulgaris diagnostic codes in EHR

| Publication | Disease | Classification | EHR | Population | Validation method | PPV (95% CI) |
|---|---|---|---|---|---|---|
| Hsu et al[21] | Pemphigus* (694.4) and bullous pemphigoid (694.5) | ICD-9 Clinical Modification diagnostic codes | Northwestern Medicine Electronic Data Warehouse; inpatient+outpatient data | Patients with multiple codes for either pemphigus (n=161) or bullous pemphigoid (n=126) | Manual review of outpatient and inpatient records. Diagnosis was confirmed by ≥2 indicators of disease: ► Clinical diagnosis ► Biopsy histology ► Direct immunofluorescence ► Indirect immunofluorescence or ELISA | Pemphigus: 100% (96% to 100%); bullous pemphigoid: 99% (93% to 99%) |
| Grönhagen et al[20] | Bullous pemphigoid (L12.0, L12.8, L12.9) | ICD-10 diagnostic codes | Swedish National Patient Register; inpatient+outpatient data | 307 patients with primary or secondary diagnosis of bullous pemphigoid who had medical records available for review | Manual review of immunopathological and histopathological registries and medical records. Diagnosis was confirmed by: ► Clear clinical history and physical examination, and ► Positive ELISA, or ► Histopathological or immunopathological pattern consistent with bullous pemphigoid | 92% |

*Pemphigus vulgaris, pemphigus vegetans, pemphigus foliaceus, pemphigus erythematosus, paraneoplastic pemphigus and drug-induced pemphigus.
EHR, electronic healthcare record; ICD-10, International Classification of Diseases Version 10; PPV, positive predictive value.

PPV for bullous pemphigoid Read codes in the CPRD is comparable to that found in routinely collected healthcare data from Sweden (National Patient Register; 92%), which validated bullous pemphigoid ICD-10 codes using information from pathology registers and medical records.[20] In contrast, the PPVs for bullous pemphigoid and pemphigus vulgaris Read codes were lower in the CPRD than those presented for the electronic medical records in the USA (Northwestern Medicine Electronic Data Warehouse; 99% for bullous pemphigoid and 100% for pemphigus ICD-9 codes).[21] However, Hsu et al[21] presented the PPV only for those cases with multiple disease codes for the same disease, and not for those with a single code or those with conflicting codes. The stringency of the case definition may explain the higher PPV found, but would not be suitable if the study aim was to identify all (or nearly all) cases in a population-based study. Furthermore, the work included several subsets of pemphigus disease, rather than just pemphigus vulgaris, and did not provide any indication of potential misclassification between different pemphigus diseases. Further comparisons between our study and the aforementioned studies are hindered as the studies validated ICD-9 and ICD-10 codes rather than Read codes. However, the

evidence indicates that routinely collected healthcare data may be a valuable source for investigating bullous pemphigoid. The role of such data sources for investigating pemphigus vulgaris remains uncertain.

### Strengths and limitations
The strengths of the present work include a large sample size and the application of algorithms to identify both primary care and benchmark hospital inpatient diagnoses. One-off occurrences of a bullous pemphigoid or pemphigus vulgaris code in the presence of alternate blistering disease codes may indicate a working diagnosis that is later revised or may be incorrect due to a typographical error or oversight. Based on clinical knowledge, using an algorithm that prioritised diagnoses with multiple entries and consistent coding was considered to help reduce the impact of recording errors.

The study is limited by the inability to calculate the negative predictive value, sensitivity and specificity of the Read codes. Additionally, HES diagnoses were regarded as the benchmark in the present study, but may be inaccurate. The benchmark diagnoses may be subject to misclassification bias due to a heterogeneous group of pemphigoid variants (eg, non-bullous pemphigoid, which

presents with intense itching, a polymorphic rash, but no blisters), alternative approaches to making a diagnosis and differences in the diagnostic tests used to confirm the diagnosis.[22 23] Nevertheless, this was regarded as the best approach as external validation with hospital notes was not possible. Finally, validation was only possible for a third of patients and it may be that patients with a hospitalisation record for a blistering disease are not representative of the overall population of patients as they are more likely to have severe disease and associated comorbidities.

### Possible explanations for findings

Almost a quarter of patients with Read codes indicative of pemphigus vulgaris in the CPRD may have bullous pemphigoid. Such misclassification may be due to unfamiliarity with the disease by GPs and administrators or typographical error. Pemphigus and pemphigoid sound similar and, to those unfamiliar with the diseases, the terms may be thought to be interchangeable. Additionally, the lower prevalence of pemphigus vulgaris in the population might explain why lower PPVs were reported for pemphigus vulgaris than bullous pemphigoid.

Inclusion of broad Read codes also contributes to the apparent misclassification seen. For example, although we have assumed that broad pemphigoid codes identify a patient with bullous pemphigoid, such a code could correctly be attributed to a patient with a different pemphigoid disease, such as mucous membrane pemphigoid. Similarly, a patient with pemphigus foliaceus could correctly be attributed a broad pemphigus code. Correct attribution of broad Read codes to patients with pemphigoid or pemphigus diseases other than bullous pemphigoid or pemphigus vulgaris likely explains the lower PPVs seen for the broad versus specific Read codes.

### Possible implications for future research

Implementation of an algorithm allows a consistent and reproducible approach to identifying patients in the presence of multiple, sometimes conflicting, blistering disease codes. The algorithm was developed based on clinical expertise and future research should be conducted to test alternate approaches and determine the sensitivity, specificity and negative predictive value.

Focused efforts on increasing GPs' awareness of autoimmune blistering skin disorders and providing resources for how to differentiate between bullous pemphigoid and pemphigus vulgaris may be required to improve the recording of the diseases in the CPRD. Currently, the discrepant codes observed complicate the examination of pemphigus vulgaris in the CPRD, and may also underestimate the true incidence of bullous pemphigoid. However, only a minor proportion of the population with a benchmark diagnosis for bullous pemphigoid (<3%) were miscoded as pemphigus vulgaris in primary care, and the effect may therefore be minimal.

## CONCLUSION

The PPVs of Read codes for bullous pemphigoid are high in the CPRD, but only moderate for pemphigus vulgaris. The CPRD is a useful data source for bullous pemphigoid research, but may have lower utility for pemphigus vulgaris. We suggest future studies of bullous pemphigoid in the CPRD should consider using this study's validated Read code list to identify cases. EHRs, such as the CPRD, could be an invaluable resource for the examination of a rare disease such as pemphigus vulgaris, but there is a need for more accurate recording of diagnoses in primary care.

**Contributors** MSMP, KEH, YV, SML, JHC, KST and SG contributed to the conception and design. MSMP, YV and SG contributed to data acquisition and analysis. MSMP, KEH, YV, SML, JHC, KST and SG contributed to the interpretation of data. MSMP drafted the manuscript and KEH, YV, SML, JHC, KST and SG critically revised the manuscript. SG is the guarantor of the work.

**Funding** This work was supported by the National Institute for Health Research (NIHR) Research for Patient Benefit grant (PB-PG-0817-20033). In addition, SML was supported by a Wellcome Senior Clinical Fellowship in Science (205039/Z/16/Z) grant.

**Disclaimer** The views expressed are those of the authors and not necessarily those of the NIHR or the Department of Health and Social Care.

**Patient consent for publication** Not required.

**Ethics approval** The present study was approved by the Independent Scientific Advisory Committee for the CPRD (ISAC protocol number 18_224).

**Provenance and peer review** Not commissioned; externally peer reviewed.

**Data availability statement** Data may be obtained from a third party and are not publicly available. Data can be requested from www.cprd.com.

**ORCID iDs**
Monica S M Persson http://orcid.org/0000-0002-8532-3006
Kim S Thomas http://orcid.org/0000-0001-7785-7465

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
