## [Reviewer comments · BMJ Open]

ARTICLE DETAILS

TITLE (PROVISIONAL)	A validation study of bullous pemphigoid and pemphigus vulgaris recording in routinely collected electronic primary healthcare records in England
AUTHORS	Persson, Monica; Harman, Karen; Vinogradova, Yana; Langan, Sinead; Hippisley-Cox, Julia; Thomas, Kim; Gran, Sonia

VERSION 1 - REVIEW

REVIEWER	J.M. Meijer University Medical Center Groningen, the Netherlands
REVIEW RETURNED	16-Dec-2019

GENERAL COMMENTS	The authors evaluated the validity of data from the Clinical Practice Research Datalink for bullous pemphigoid and pemphigus vulgaris and whether these data can be used to study the blistering diseases. The study provides a first insight in the conformity between diagnosis of bullous diseases in primary health care and of hospital diagnosis. The authors conclude the CPRD codes can be used to study bullous pemphigoid, but are not suitable to study pemphigus vulgaris. The manuscript provides a balanced discussion of the value of CPRD codes and potential misclassification of bullous diseases. Comments 1. The manuscript is clear and reads well, but it may also need the complete introduction and methods to understand the discussion of results. The authors might consider to explain the specific and broad Read codes in short in the discussion section, page 11 line 23-27.2. The study involved a large sample size of primary health care cases, although only approximately 30% could be used to compare with hospital inpatient diagnosis. This can be emphasized more in the results, in abstract page 3 line 35 and page 9 line 15.3. Figure 3 shows the distribution of age at diagnosis of bullous pemphigoid and pemphigus vulgaris of the CPRD diagnosis. In the Discussion section page 11 line 48 the authors comment the probable misdiagnosis of bullous pemphigoid in the pemphigus vulgaris group likely explains the peak in age group 75 to 79 years, corresponding the bullous pemphigoid peak. This important point not clear at first sight in the figure legenda. Although a smaller sample size, more appropriate would be a figure of the age distribution of validated diagnosis.
---

	4. The reported misclassification of diagnosis of bullous diseases prompts the question how diagnosis is made in the hospital setting. A potential bias for misclassification is the lack of diagnostic criteria for bullous pemphigoid, a heterogeneous group of pemphigoid variants and differences in used diagnostic tests to confirm diagnosis. This would be of additional value in the discussion, with suggested reference to Smith et al. National audit on the management of bullous pemphigoid. Clinical and Experimental Dermatology, august 2019. Doi:10.1111/ced.14086
--	--

REVIEWER	Roberto Maglie Department of Health Sciences, Section of Dermatology, University of Florence
REVIEW RETURNED	02-Jan-2020

GENERAL COMMENTS	Interesting article that investigate the validity of BP and PV recording in a healthcare data provided by GP in UK. Data about epidemiological impact of such disease in the UK population, especially for BP, are in line with previous epidemiological data. I have only two minor observation. 1) Since some PV cases were subsequently diagnosed as BP, one implicating finding of this study is to increase awareness among GP about the key differences between these 2 blistering dermatosis, e.g. early and often devastating mucosal involvement as a diagnostic clue of PV (mucosal involvement is rarely seen in BP); itching as a diagnostic clue of BP. 2) I would consider within the limitation of the study a possible underestimation of BP due to the only recently recognized BP subtypes characterized by the absence of blisters, referred to as non-bullous pemphigoid. BP (or pemphigoid) may presents as itching without clinically visible lesions, which makes challenging its recognition until patient's referral to the dermatologist and after performing direct immunofluorescence to detect an antibody-based autoimmune process.
--

REVIEWER	Kyle Amber University of Illinois at Chicago, USA
REVIEW RETURNED	06-Jan-2020

GENERAL COMMENTS	This is an excellent study. Studies validating database diagnoses allow confidence in future research utilizing these databases. The authors' methodology was spot on to validate this (particularly in BP).
--

REVIEWER	SCAILTEUX Lucie-Marie Clinical Pharmacology Department Unit of Pharmacovigilance and Pharmacoepidemiology
REVIEW RETURNED	17-Feb-2020

GENERAL COMMENTS	In this study, authors assessed the external validity of bullous pemphigoid and pemphigus vulgaris Red codes through the linkage
--

	between CPRD (a primary care database) and HES database (secondary care database). Authors had a well-structured study and well-written manuscript. They also recognized their limitations of the use of ICD-10 codes without access to clinical data to confirm the diagnoses, which is most important limitation of the study in my opinion. Minor comments: - page 4, line 39: authors suggest that the prescription data allows association between drug exposure and the bullous pemphigoid. Authors should specify what kind of treatment are used to treat the disease, if the drug indication is available in the database, and if these treatments are specific of the disease. Without information on the true indication, the use of a treatment (oral corticosteroid for instance) is not a sufficient argument to specify the treatment indication. A concomitant disease can be present. In addition, authors should specify how these drug exposure data were used considering the purpose of the study. Major comments: - authors should specify how is performed the databases linkage and if there is a risk of multiple identities in the hospital database for each patient identified in the CPRD database (risk of misclassification). - page 7, line 49: authors indicate that all other blistering ICD-10 codes and where the HES algorithm generated uncertain diagnoses were classed as "other". How were considered the cases with the Read codes not corresponding to an ICD-10 code (M151.00 Erythema multiforme, M151.12 Toxic epidermal necrolysis, for instance)? Were they included in the "other" group? - regarding the Supplementary Table 1, some pemphigus codes have been classified in the group "others". Why didn't you group them together with the pemphigus codes? In the group "Others", code L13.1 'Subcorneal pustular dermatitis' should not be considered as a disease related to a bullous pemphigoid or pemphigus, as it is a different disease. In general, more details and explanations should be provided for the "Others" group, as we do not really understand which codes are taken into account or not. - authors should provide a flowchart showing the databases recovering. In addition, how were considered the patients with bullous pemphigoid and pemphigus vulgaris Read codes in CPRD who were not hospitalised? An underestimation of number of cases could happen. It should be more discussed by authors. - A possible classification bias is present, considering that some patients might not have been correctly identified with blisters, treated only at the GP level, and not hospitalized (therefore not found in the HES database) when they actually had a bullous pemphigoid or a pemphigus vulgaris (problem of low sensitivity). It should be more discussed by authors.
--	--

VERSION 1 – AUTHOR RESPONSE

REVIEWER 1 – Joost M. Meijer (University Medical Center Groningen)	
General	The authors evaluated the validity of data from the Clinical Practice Research Datalink for bullous pemphigoid and pemphigus vulgaris and whether these data can be used to study the blistering diseases. The study provides a first insight in the conformity between diagnosis of bullous diseases in primary health care and of

	hospital diagnosis. The authors conclude the CPRD codes can be used to study bullous pemphigoid, but are not suitable to study pemphigus vulgaris. The manuscript provides a balanced discussion of the value of CPRD codes and potential misclassification of bullous diseases.
Comment 1	The manuscript is clear and reads well, but it may also need the complete introduction and methods to understand the discussion of results. The authors might consider to explain the specific and broad Read codes in short in the discussion section, page 11 line 23-27.
Response 1	Thank you for this interesting point. We have added a statement in the beginning of the discussion to explain the specific versus broad Read codes. We hope that this will allow readers to better understand the points raised in the discussion without having to refer back to the methods.
Change 1	Discussion, page 11, Lines 24-27: “Diagnoses were recorded in the CPRD using specific (M145000, “bullous pemphigoid”; M144600, “pemphigus vulgaris”; M144500, “pemphigus vegetans”) or broad (M145.00, “pemphigoid”; M145z00, “pemphigoid not otherwise specified”; M144.00, “pemphigus”; M144z00, “pemphigus not otherwise specified”) Read codes.”
Comment 2	The study involved a large sample size of primary health care cases, although only approximately 30% could be used to compare with hospital inpatient diagnosis. This can be emphasized more in the results, in abstract page 3 line 35 and page 9 line 15.
Response 2	We agree that readers would benefit from having this fact better emphasised throughout. We have done this by: 1) adding the percentages that had hospital inpatient diagnoses to the abstract, 2) adding a statement in the results that explicitly states that the 70% without an inpatient diagnosis were not included in the validation study, and 3) showing the numbers of patients analysed in the flowchart that has been added to the supplementary material.
Change 2	Abstract, page 3, line 16-18: “Of 2,468 incident cases of bullous pemphigoid and 431 of pemphigus vulgaris, 797 (32.3%) and 85 (19.7%) patients, respectively, had a hospitalisation record for a blistering disease.” Results, page 9, lines 13-14: “The remaining 2,017 (69.6%) patients did not have any hospitalisation records listing a blistering disease diagnosis and were not analysed in the validation study.” Addition of supplementary figure 1
Comment 3	Figure 3 shows the distribution of age at diagnosis of bullous pemphigoid and pemphigus vulgaris of the CPRD diagnosis. In the Discussion section page 11 line 48 the authors comment the probable misdiagnosis of bullous pemphigoid in the pemphigus vulgaris group likely explains the peak in age group 75 to 79 years, corresponding the bullous pemphigoid peak. This important point not clear at first sight in the figure legenda. Although a smaller sample size, more appropriate would be a figure of the age distribution of validated diagnosis.
Response 3	We had previously described the two peaks observed in the results section. Following the reviewer’s suggestion, we have now also emphasised this finding in the figure legend. We feel it is important to show the age distribution for all the CPRD as this provides important evidence for our conclusion that a significant proportion of patients with pemphigus vulgaris codes in the CPRD actually have bullous pemphigoid. In the introduction (page 4 lines 27-28 to page 5 lines 1-4) we refer to the age distributions of the paper by Langan et al (2008) as the basis for conducting our work – the peak age of patients with pemphigus vulgaris in their electronic health records was also high and indicated a possible misclassification. By presenting the age distributions for all cases, we are able to provide data that directly compares with the earlier

	publication. As the reviewer mentioned in comment 2, we have only been able to analyse a subset of patients (30%) who have a blistering disease diagnosis available in the CPRD and HES. Ideally we would have liked to examine the diagnoses of all patients, but this has not been possible. We feel that the best way to maximise the information we have available is therefore to present the age distributions of all patients. This is the best evidence we have available from all CPRD patients that gives an indication of whether or not the diagnosis is correct.
Change 3	Figure 3 legend: "Figure 3 - Age at diagnosis (years) in patients with bullous pemphigoid (n=2,468) and pemphigus vulgaris (n=431). A peak is observed at 80-84 years for bullous pemphigoid recording, whilst two peaks are seen for pemphigus vulgaris; one at 50-54 and a later, and slightly higher, peak at 75-79 years."
Comment 4	The reported misclassification of diagnosis of bullous diseases prompts the question how diagnosis is made in the hospital setting. A potential bias for misclassification is the lack of diagnostic criteria for bullous pemphigoid, a heterogeneous group of pemphigoid variants and differences in used diagnostic tests to confirm diagnosis. This would be of additional value in the discussion, with suggested reference to Smith et al. National audit on the management of bullous pemphigoid. Clinical and Experimental Dermatology, august 2019. Doi:10.1111/ced.14086
Response 4	Thank you for this insightful suggestion and reference. This, along with a systematic review on non-bullous pemphigoid, has been incorporated into the discussion
Change 4	Discussion, page 15, lines 1-5: "The benchmark diagnoses may be subject to misclassification bias due to a heterogeneous group of pemphigoid variants (e.g., non-bullous pemphigoid, which presents with intense itching, a polymorphic rash, but no blisters), alternative approaches to making a diagnosis, and differences in the diagnostic tests used to confirm the diagnosis. ^{22,23} "
REVIEWER 2 – Roberto Maglie (University of Florence)	
General	Interesting article that investigate the validity of BP and PV recording in a healthcare data provided by GP in UK. Data about epidemiological impact of such disease in the UK population, especially for BP, are in line with previous epidemiological data.
Comment 1	Since some PV cases were subsequently diagnosed as BP, one implicating finding of this study is to increase awareness among GP about the key differences between these 2 blistering dermatosis, e.g. early and often devastating mucosal involvement as a diagnostic clue of PV (mucosal involvement is rarely seen in BP); itching as a diagnostic clue of BP.
Response 1	We agree that there is a need for GPs to have a greater awareness of the diseases as this would not only improve patient care, but also facilitate future research using the CPRD. We have incorporated the need for GP education into the discussion
Change 1	Discussion, page 16, lines 7-10: "Focused efforts on increasing GPs' awareness of autoimmune blistering skin disorders and providing resources for how to differentiate between bullous pemphigoid and pemphigus vulgaris may be required to improve the recording of the diseases in the CPRD."
Comment 2	I would consider within the limitation of the study a possible underestimation of BP due to the only recently recognized BP subtypes characterized by the absence of blisters, referred to as non-bullous pemphigoid. BP (or pemphigoid) may presents as itching without clinically visible lesions, which makes challenging its recognition until patient's referral to the dermatologist and after performing direct immunofluorescence to detect an antibody-based autoimmune process.
Response 2	Thank you for this suggestion. We agree that a limitation of the work is that diagnoses of bullous pemphigoid may be missed because of different variants (e.g., bullous versus non-bullous pemphigoid). We have incorporated this into a newly

	added limitation regarding misclassification bias. It may be that cases are missed because of unfamiliarity with this newly recognised subtype, therefore contributing to misclassification.
Change 2	Discussion, page 15, lines 1-5: “The benchmark diagnoses may be subject to misclassification bias due to a heterogeneous group of pemphigoid variants (e.g., non-bullous pemphigoid, which presents with intense itching, a polymorphic rash, but no blisters), alternative approaches to making a diagnosis, and differences in the diagnostic tests used to confirm the diagnosis. ^{22,23} ”
REVIEWER 3 – Kyle Amber (University of Illinois)	
General	This is an excellent study. Studies validating database diagnoses allow confidence in future research utilizing these databases. The authors' methodology was spot on to validate this (particularly in BP).
REVIEWER 4 – Lucie-Marie Scailteux (Unit of Pharmacovigilance and Pharmacoepidemiology)	
General	In this study, authors assessed the external validity of bullous pemphigoid and pemphigus vulgaris Red codes through the linkage between CPRD (a primary care database) and HES database (secondary care database). Authors had a well-structured study and well-written manuscript. They also recognized their limitations of the use of ICD-10 codes without access to clinical data to confirm the diagnoses, which is most important limitation of the study in my opinion
Comment 1	(MINOR) page 4, line 39: authors suggest that the prescription data allows association between drug exposure and the bullous pemphigoid. Authors should specify what kind of treatment are used to treat the disease, if the drug indication is available in the database, and if these treatments are specific of the disease. Without information on the true indication, the use of a treatment (oral corticosteroid for instance) is not a sufficient argument to specify the treatment indication. A concomitant disease can be present. In addition, authors should specify how these drug exposure data were used considering the purpose of the study.
Response 1	Thank you for this comment. Our aim with the statement regarding prescription data was to give the reader an overview of the CPRD and illustrate the potential uses of the database in relation to bullous pemphigoid. We give the example of drug exposure as this is an area of current interest in the field of bullous pemphigoid which could be addressed using CPRD. We also explain that longitudinal follow-up allows long term outcomes to be determined (e.g., mortality), thus highlighting another example for how CPRD data could be used. We aimed to show the importance of our work and to justify the reasons for conducting it – our work forms the first step, from which further research into bullous pemphigoid can be based. We have not used prescription data in the present study. Although the reviewer raises some valid points for discussion where prescription data are used, these are not within the aims or scope of the present work. We will take these helpful comments on board, however, when we plan future work using the prescription data in the CPRD.
Comment 2	Authors should specify how is performed the databases linkage and if there is a risk of multiple identities in the hospital database for each patient identified in the CPRD database (risk of misclassification).
Response 2	Linkage of the databases is performed by a trusted third party (NHS Digital), who uses a step-wise deterministic approach based on a number of unique and partial identifiers. The NHS number, which is unique to each person and remains unchanged throughout their lifetime, is the main identifier used when performing linkages. For over 96% of people in the CPRD, linkage occurs in the first two steps of match ranking (exact NHS number, gender, date of birth, +/- postcode). We believe

	the risk of multiple identities is limited and unlikely to significantly impact the findings. We have summarised the linkage process in the methods and provided a reference to a publication which gives a more detailed explanation of the linkage.
Change 2	Methods, page 8, lines 6-8: "Linkage is based on each patient's NHS number, which is unique and remains unchanged through their lifetime, along with other identifiers (e.g., gender, date of birth, postcode).¹⁶"
Comment 3	page 7, line 49: authors indicate that all other blistering ICD-10 codes and where the HES algorithm generated uncertain diagnoses were classed as "other". How were considered the cases with the Read codes not corresponding to an ICD-10 code (M151.00 Erythema multiforme, M151.12 Toxic epidermal necrolysis, for instance)? Were they included in the "other" group?
Response 3	For the present study, our focus was on diagnoses for bullous pemphigoid and pemphigus vulgaris. The "other" indicates any blistering disease diagnosis that is not bullous pemphigoid or pemphigus vulgaris. We analysed the patients according to the three categories: bullous pemphigoid, pemphigus vulgaris, and "other". We did not consider the "other" category in any more detail and did not directly match Read and ICD-10 codes within this category. The validation study only analysed patients with an overall CPRD diagnosis of bullous pemphigoid or pemphigus vulgaris – those where the algorithm generated an "other blistering disease" diagnosis were not considered cases. This is described in the case definition. We have updated the methods to clarify our case definition.
Change 3	Methods, page 7, lines 2-6: "Patients were considered cases if their most likely blistering disease diagnosis, generated through the algorithm, was bullous pemphigoid or pemphigus vulgaris. Those with diagnoses for other blistering diseases were excluded. Further information about the blistering disease codes used is available in Supplementary Table 1."
Comment 4	regarding the Supplementary Table 1, some pemphigus codes have been classified in the group "others". Why didn't you group them together with the pemphigus codes? In the group "Others", code L13.1 'Subcorneal pustular dermatitis' should not be considered as a disease related to a bullous pemphigoid or pemphigus, as it is a different disease. In general, more details and explanations should be provided for the "Others" group, as we do not really understand which codes are taken into account or not.
Response 4	Thank you for raising these points. We can see that this is an area that could cause readers some confusion, so have provided more information in the supplementary information. With regards to each of the points raised: We didn't group all pemphigus codes based on clinical opinion - we are interested in pemphigus vulgaris and not other pemphigus diseases. For example, pemphigus foliaceus is a different clinical entity, so was not classed as pemphigus vulgaris. "Other pemphigus" is a code used where the patient has a pemphigus disease that is not covered by named diseases that have specific codes (e.g., IgA pemphigus). We have addressed the fact that our broad codes might cover these other types of pemphigus as a limitation. Regarding Read codes (primary care): The Read code lists were generated using clinical expertise, where the aim was to generate a list of any disease that could potentially be differential diagnoses for bullous pemphigoid or pemphigus vulgaris. As described in the methods, we identified all patients with at least one code for bullous pemphigoid or pemphigus vulgaris. However, we felt that some people with a different blistering disease may have a one-off code for bullous pemphigoid or pemphigus vulgaris, perhaps whilst it is a working diagnosis that is later proven incorrect. We therefore applied the algorithm to all blistering disease codes and have

	provided the list of codes that were considered in the supplementary table 1. ICD-10 codes (secondary care): We have described in the methods how we identified patients with a “blistering disease” according to the broad ICD headings – L10 (pemphigus), L12 (pemphigoid), L13 (other bullous disorders). These were the disease codes we felt patients with bullous pemphigoid or pemphigus vulgaris would most likely be misclassified as having. We applied the algorithm to determine the most likely blistering disease diagnosis using HES data, as described. In the methods (page 7, lines 26-28), we describe which ICD codes correspond to bullous pemphigoid and pemphigus vulgaris. The “other” category can therefore be regarded as “not having bullous pemphigoid or pemphigus vulgaris”. We agree that “subcorneal pustular dermatitis” is not bullous pemphigoid or pemphigus vulgaris, hence why it is classed as “Other”. However, we feel that the differential diagnosis for a patient with blisters may include this disease, so there could be misclassification occurring between bullous pemphigoid/pemphigus vulgaris and subcorneal pustular dermatitis.
Change 4	Methods, page 7, lines 2-6: “Further information about the blistering disease codes used is available in Supplementary Table 1.” Supporting information: “The CPRD and HES algorithms used to determine the most likely blistering disease diagnosis were run using all entries for the blistering diseases described below. Based on clinical expertise, we determined Read and ICD-10 codes for bullous pemphigoid and pemphigus vulgaris. We also generated a list of “other blistering diseases” which could be confused with or be differential diagnoses for bullous pemphigoid or pemphigus vulgaris.”
Comment 5	authors should provide a flowchart showing the databases recovering. In addition, how were considered the patients with bullous pemphigoid and pemphigus vulgaris Read codes in CPRD who were not hospitalised? An underestimation of number of cases could happen. It should be more discussed by authors.
Response 5	Regarding those that did not have a hospital code for a blistering disease, these patients were excluded from the validation study – this has now been clearly stated in the results. In the discussion, we highlight this as a limitation (please see our response to comment 6 for further detail). We have added patient flow diagram in the supplementary information which shows the identification of patients from the CPRD. In it, we have also clearly indicated the numbers that were included in the validation study.
Change 5	Results, page 9, lines 13-14: “The remaining 2,017 (69.6%) patients did not have any hospitalisation records listing a blistering disease diagnosis and were not analysed.” Results, page 8 lines 27-28 to page 9 line 1: “A total of 2,899 incident cases of bullous pemphigoid and pemphigus vulgaris were identified in HES-linked practices using the algorithm (see Supplementary Figure 1 for patient flow diagram).” Supplementary figure 1 added: “Flow diagram showing identification of incident cases of bullous pemphigoid and pemphigus vulgaris for inclusion in the validation study”
Comment 6	A possible classification bias is present, considering that some patients might not have been correctly identified with blisters, treated only at the GP level, and not hospitalized (therefore not found in the HES database) when they actually had a bullous pemphigoid or a pemphigus vulgaris (problem of low sensitivity). It should be more discussed by authors.
Response 6	We agree that this is a limitation of our work and have listed this in the discussion: “Finally, validation was only possible for a third of patients and it may be that patients with a hospitalisation record for a blistering disease are not representative of the overall population of patients as they are more likely to have severe disease and associated comorbidities”.

VERSION 2 – REVIEW

REVIEWER	Joost M. Meijer Center for Blistering Disease, Department of Dermatology, University Medical Center Groningen
REVIEW RETURNED	23-Mar-2020

GENERAL COMMENTS	Well balanced article with discussion of new insights and limitations of the study and with the comments of reviewers addressed.
--

REVIEWER	Roberto Maglie Department of Health Sciences, Section of Dermatology, University of Florence
REVIEW RETURNED	20-May-2020

GENERAL COMMENTS	thanks for making the suggested revisions. i have no further comments.
---

REVIEWER	SCAILTEUX Lucie-Marie Rennes University Hospital, Unit of Pharmacovigilance and Pharmacoepidemiology, France
REVIEW RETURNED	19-Mar-2020

GENERAL COMMENTS	Authors took into consideration all my comments and brought relevant responses. Recommandation to accept the manuscript.
--